# Evaluation of 3D Ultrasound Reconstruction and 2D/3D Segmentation for Neonatal Hip Dysplasia Screening

**Wiebke Heyer**[1] (iD)                                   W.HEYER@UNI-LUEBECK.DE
**Katharina Ott**[1]                        KATHARINA.OTT@STUDENT.UNI-LUEBECK.DE
[1] *University of Lübeck, Institute of Medical Informatics, Ratzeburger Allee 160, Lübeck, Germany*
**Christian Weihsbach**[2] (iD)                           CHRISTIAN@ECHOSCOUT.AI
[2] *Echoscout GmbH, Maria-Goeppert-Str. 3, Lübeck, Germany*
**Reza Sorbi**[3] (iD)                                       REZA.SORBI@UKSH.DE
**Lisa Lange**[3]                                            LISA.LANGE@UKSH.DE
[3] *Universitätsklinikum Schleswig-Holstein, Campus Kiel, Kiel, Germany*
**Jürgen Lichtenstein**[2]                               JUERGEN@ECHOSCOUT.AI
**Anna Hell**[4] (iD)                            ANNA.HELL@MED.UNI-GOETTINGEN.DE
[4] *Universitätsmedizin Göttingen, Robert-Koch-Straße 40, Göttingen, Germany*
**Sebastian Lippross**[5] (iD)                        SLIPPROSS@SCHOEN-KLINIK.DE
[5] *Schön Klinik, Lilienstraße 20-28, Rendsburg, Germany*
**Lasse Hansen**[2] (iD)                                     LASSE@ECHOSCOUT.AI
**Mattias P. Heinrich**[1] (iD)                 MATTIAS.HEINRICH@UNI-LUEBECK.DE

**Editors:** Accepted for publication at MIDL 2026

## Abstract

Early detection of developmental dysplasia of the hip relies heavily on the correct acquisition and interpretation of ultrasound images. Yet, conventional single-plane imaging provides only a limited view of the neonatal hip, is operator-dependent and sensitive to probe orientation. In this study, we present a clinically oriented validation of a dual-sweep 3D ultrasound approach aimed at improving anatomical coverage and simplifying the diagnostic process. Our dataset comprises 50 optically tracked acquisitions and 150 untracked freehand sweeps from newborns, enabling the reconstruction of volumetric representations of the hip from standard handheld 2D ultrasound. We evaluate 2D and 3D nnU-Net–based segmentation models to quantify how volumetric context influences the delineation of key joint structures. Results demonstrate that the combination of 2D slice-based and 3D volumetric segmentation yields the most robust performance, particularly in cases with anatomical variability or suboptimal sweep direction. The study also highlights remaining challenges, including motion artefacts and inconsistent sweep trajectories, that affect reconstruction quality.

**Keywords:** Hip Dysplasia, Ultrasound, Validation, Segmentation, 3D Reconstruction.

## 1. Introduction

Developmental dysplasia of the hip (DDH) is one of the most common musculoskeletal disorders in neonates, encompassing a spectrum of hip joint abnormalities ranging from mild acetabular dysplasia to complete dislocation of the femoral head. Early diagnosis and intervention are critical, as untreated DDH can lead to long-term complications and the

need for complex surgical procedures in later life. Despite the existence of well-established screening programs in many countries a subset of cases remains undetected until adolescence or adulthood, highlighting the limitations of current diagnostic practices. Traditional DDH screening relies primarily on two-dimensional (2D) ultrasonography, most commonly performed using the Graf method(Graf, 1980). This approach requires precise identification of standardized planes through the hip joint and manual angle measurement to assess bony and cartilaginous coverage of the femoral head. While widely adopted, 2D ultrasound is inherently operator-dependent and provides only a planar view of a complex 3D anatomy. Variability in probe alignment, infant positioning, and interpretation can lead to inconsistent diagnoses.

Recent advances in deep learning (DL) have enabled the reconstruction of 3D volumes from freehand 2D ultrasound sweeps, offering a potential solution to the limitations of conventional 2D imaging. By capturing volumetric information, these methods provide a more comprehensive representation of the hip morphology. Furthermore, 3D reconstruction facilitates retrospective analysis, may reduce inter-observer variability, and lays the groundwork for learning-based diagnostic support to improve screening accuracy and consistency.

In this study, we explore the combination of 2D- and 3D-deep-learning techniques to enhance neonatal hip assessment. We introduce a real-world clinical dataset of neonatal hip ultrasounds, including both Graf-standard 2D sweeps and freehand sweeps suitable for volumetric reconstruction. Using this dataset, we evaluate a staged segmentation approach, adapt a freehand 3D reconstruction technique, and establish a complete workflow from data acquisition to 3D hip modeling.

## 2. Background

### 2.1. Developmental Dysplasia of the Hip

Developmental dysplasia of the hip encompasses a spectrum of congenital abnormalities affecting the morphology and stability of the hip joint, ranging from mild acetabular dysplasia to complete dislocation of the femoral head (Gkiatas et al., 2019). It is one of the most common congenital musculoskeletal disorders, with an incidence of 1–7 % in newborns, markedly higher in girls (Laskaratou et al., 2024; Gkiatas et al., 2019). Epidemiology varies widely across populations, from as low as 0.06 cases per 1,000 live births in African populations to over 70 per 1,000 among Native American groups (Loder and Skopelja, 2011). Established risk factors include female sex, breech presentation, positive family history, and first birth among others (Loder and Skopelja, 2011). In the healthy neonatal hip, the femoral head is stably seated within the cartilaginous acetabulum. In DDH, insufficient acetabular depth or coverage results in joint laxity, causing Ortolani sign (Ortolani, 1976), slippage of the femoral head over the acetabular rim. The acetabulum shows reduced depth, altered orientation and shape changes while retaining its overall width. Hence, femoral head coverage is inadequate, the centre of rotation shifts laterally, and the diminished contact area leads to asymmetric loading and cartilage or labral damage. Clinically, untreated DDH can result in limping, gait abnormalities, pain, early degenerative changes, and, in severe cases, hip dislocation. It is a major risk factor for early-onset hip osteoarthritis, identified in 20–40 % of patients with degenerative hip disease (Gala et al., 2016). Early detection is critical: when diagnosed during the neonatal period, conservative treatment, most commonly with

a Pavlik harness, successfully normalises the joint in the majority of infants. In contrast, cases detected later in infancy or childhood often require more invasive interventions, such as complex reconstructive surgery or, ultimately, total hip replacement in adulthood. Despite established screening pathways, a substantial proportion of cases are only identified during adolescence or adulthood; in the US' adult population, the prevalence is estimated at approximately 0.1 % (Gala et al., 2016).

## 2.2. Diagnosis of DDH

The diagnosis of DDH in early infancy relies on clinical examination and ultrasonographic imaging (Graf, 1980). In Germany, universal ultrasound screening is mandated during routine examinations in the first weeks of life, typically performed before hospital discharge or shortly thereafter (Seidl et al., 2024). In contrast, some countries, such as the United States, reserve ultrasound for infants with clinical findings or known risk factors (Shaw et al., 2016). Ultrasound is the preferred modality due to its accessibility, low cost, safety, and real-time dynamic bedside imaging. Standard 2D ultrasound is sufficient for routine assessment but is highly operator-dependent, requiring precise identification of an anatomical plane and reliable visualisation of key landmarks. Since the early 1980s, the Graf method has been the gold standard in many European countries (Graf, 1980). It classifies hips by quantifying bony and cartilaginous femoral head coverage. A valid exam requires a standardised coronal plane with a sharply defined lower iliac bone. The infant lies in lateral decubitus, and the transducer is aligned to visualise key structures, including the chondro-osseous junction, femoral head, capsule, labrum, cartilage, and acetabular roof. Two lines define acetabular morphology: the baseline along the lateral iliac border and the bony roof line. Their $\alpha$-angle measures bony coverage; the $\beta$-angle with the cartilage roof line reflects cartilaginous coverage. These angles underpin the Graf classification: Type I ($\alpha \geq 60°$) is normal; Type II ($43° \leq \alpha \leq 59°$) reflects immaturity or dysplasia; Types III–IV (low $\alpha$ values) indicate subluxation or dislocation requiring urgent treatment. Small plane or landmark errors can cause misclassification. Although ultrasound is operator-dependent, affected by infant movement and artifacts, and lacks 3D context, it remains the most practical and widely used modality for early DDH diagnosis and screening.

## 2.3. Rationale for 3D Hip Modelling

Conventional DDH assessment relies on a single 2D ultrasound slice, even though the infant hip is a complex three-dimensional structure whose diagnostic features may not be fully captured in a planar view. The Graf method requires identifying an anatomically precise coronal plane, but this fixed perspective inevitably reduces the available spatial information to a small cross-section of the acetabulum and femoral head. Subtle abnormalities of acetabular shape, orientation and global coverage may therefore remain undetected, and the diagnostic outcome can depend heavily on the examiner's ability to align the probe with high accuracy. These limitations motivate the reconstruction of a three-dimensional representation of the hip from freehand 2D ultrasound sequences. A reconstructed 3D volume enables a much more comprehensive assessment of hip morphology. By integrating multiple partially overlapping 2D frames, the full contour of the acetabular rim, its depth and curvature, and the spatial relationship to the femoral head can be visualised rather than inferred

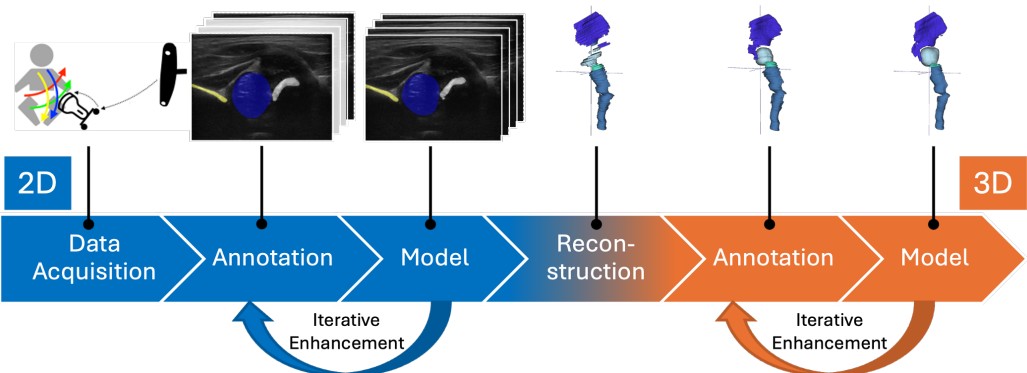

Figure 1: Overview of our staged segmentation approach.

from a single slice. This allows diagnostic decisions to be based on the global acetabular geometry rather than one predefined plane. Moreover, 3D volumes enable clinicians to retrospectively generate arbitrary slices beyond Graf's plane, analyse angles and distances from different perspectives, and explore anatomical variations not visible in standard 2D imaging.

Additionally, geometric descriptors such as volumetric femoral head coverage and acetabular depth can be computed directly from the reconstructed anatomy. These measures have the potential to improve diagnostic precision, reduce inter-observer variability, and capture structural characteristics of dysplasia that extend beyond the $\alpha$- and $\beta$-angles. Because all measurements can be performed retrospectively on the same reconstructed volume, reproducibility is increased and second opinions become more consistent.

Finally, a 3D representation provides a richer basis for machine-learning-based diagnostic support. Models can learn from the full spatial context rather than isolated 2D slices, enabling more robust detection of dysplastic patterns and more reliable estimation of clinically relevant parameters. In this way, freehand 3D ultrasound reconstruction has the potential to extend the diagnostic capabilities of widely available 2D systems and to provide a more complete and objective characterisation of infant hip morphology.

## 3. Contribution

This work presents a comprehensive investigation into the use of 2D and 3D deep learning methods to improve the diagnostic pipeline for developmental dysplasia of the hip. An overview is given in Fig. 1 and our contributions are summarised as follows:

- **Introduction of a new real-world neonatal hip ultrasound dataset.** We present a curated dataset combining Graf-standard 2D clinical ultrasound with freehand sweeps suitable for 3D reconstruction. The dataset includes high-quality semi-manual annotations of clinically relevant anatomical structures and is, to our knowledge, the first of its kind in this diagnostic domain.

- **Iterative refinement for robust segmentation.** We systematically evaluate multiple 2D and 3D nnU-Net configurations on neonatal hip ultrasound and show that

established models initially struggle on this modality. To mitigate this, we incorporate a staged refinement in which initial 2D-derived 3D masks are iteratively improved using successive nnU-Net predictions. This cascade leads to substantially higher segmentation quality while reducing the manual annotation effort required for producing anatomically coherent 3D ground truth.

- **Adaptation and validation of freehand ultrasound reconstruction for DDH.** We demonstrate that existing freehand reconstruction methods can be adapted to neonatal hip ultrasound sweeps with minimal modifications, enabling accurate 3D reconstructions despite probe trajectory variability and common acquisition artefacts.

- **End-to-end workflow from acquisition to 3D hip model.** We provide a complete workflow beginning with neonatal hip ultrasound acquisition, continuing through 2D segmentation and 3D reconstruction, and culminating in volumetric models of hip morphology suitable for further quantitative and diagnostic analysis.

## 4. Related Work

### 4.1. Hip Ultrasound Data and Segmentation

Ultrasound segmentation has seen rapid progress with deep learning also by an increasing availability of open-access datasets. Recent foundation-models, such as UltraSam by Meyer et al. (2025), leverage large multi-anatomy collections to learn modality-specific priors across 2D segmentation tasks that address the inherent challenges (speckle noise, shadows, low tissue contrast) of ultrasound acquisition. However, these are almost exclusively 2D datasets and corresponding models are not volumetrically consistent, which is an important requirement for the anatomies in hip dysplasia 3D morphology assessment. Earlier DL work has focused on improving segmentation quality of tissue boundaries in individual frames (Mishra et al., 2018) through boundary-attention mechanism. While effective for ambiguous tissue interfaces in 2D, such methods are limited by frame-by-frame training and do not capture the spatial continuity or coherence of 3D representations.

Within the domain of hip dysplasia, research on ultrasound 3D segmentation for DDH diagnosis remains limited. Initial works focused on landmark detection in 2D ultrasound images as a first step to support the diagnostic workflow in clinics (Xu et al., 2021). Consecutive works automatically calculate the Graf angles from the detected landmarks (Hu et al., 2021; Chen et al., 2024). Regarding segmentation, U-Net-based CNNs were used to automatically delineate the ilium, femoral head, and labrum in 2D Graf slices to calculate the femoral head coverage and classify DDH (Stamper et al., 2023). Hsu et al. (2025) skipped intermediate landmark detection or segmentation steps and classified DDH directly, visualizing the results with attention heatmaps to make the automated classification interpretable to clinicians.

3D freehand sweeps were considered by Hareendranathan et al. (2016), who proposed a semi-automatic workflow for segmenting echogenic structures, demonstrating the feasibility of volumetric DDH analysis but requiring substantial manual input and lacking data-driven robustness. More recently, Liu et al. presented NHBS-Net for 2D neonatal hip bone segmentation, showing good performance on slice-level datasets yet still constrained by the inherent variability and operator dependency of 2D acquisition (Liu et al., 2021).

### 4.2. Tracking-free 3D Reconstruction from Freehand 2D Ultrasound Sweeps

Tracking-free 3D reconstruction of freehand ultrasound provides a cost-effective alternative to dedicated 3D systems, balancing the limited spatial context of conventional 2D imaging with the complexity and cost of tracked setups. By reconstructing volumetric data from standard 2D sweeps without external hardware, these methods eliminate the need for calibration and avoid limitations of optical or electromagnetic tracking, such as line-of-sight requirements or sensitivity to environmental interference. This simplified workflow is particularly attractive for clinical adoption and resource-constrained settings.

A comprehensive review of the state of art in tracker-free ultrasound reconstruction is presented in (Adriaans et al., 2024). Early deep-learning approaches by Prevost et al. (2018) used feed-forward convolutional neural networks with optical flow and data from the ultrasound probes inertial measurement unit (IMU). El Hadramy et al. (2023) extended this with a siamese ConvLSTM and gaussian heatmaps to focus on relevant motion regions. Both approaches improved distance and drift errors, although drift still scaled with sequence length and angle estimation remained challenging. Guo et al. (2022) introduced contextual learning on short sequences and contrastive learning with a margin ranking loss to further reduce drift.

## 5. Material and Methods

### 5.1. Dataset and Acquisition

We collected a real-world clinical hip ultrasound dataset as part of routine neonatal care at two centres: the Universitätsklinikum Schleswig-Holstein (UKSH), Campus Kiel, and the Universitätsmedizin Göttingen (UMG). All recordings were acquired in addition to the standard Graf-based screening examination and followed the same clinical scanning protocol used for routine diagnostics as well as additional sweeps.

Ultrasound sweeps were acquired with a portable Clarius L7HD3 linear probe by trained medical staff, including experienced physicians familiar with the Graf method. For each infant, six freehand sweeps were recorded (three per hip). One sweep followed the standardised coronal plane used in the Graf classification, ensuring comparability with the clinical gold standard. The two additional sweeps were designed to provide volumetric coverage: one was acquired from the posterior gluteal region toward the hip, and the other from the pelvis across the acetabular region toward the proximal femur. These two trajectories are approximately orthogonal. Although operators were instructed to sweep at a steady and uniform velocity to obtain consistent spatial sampling, the posterior–anterior sweeps were found to be longer in duration (7.5 s on average) but cover a shorter physical path, resulting in denser image information compared to proximal–distal sweeps, which traverse a larger distance more rapidly (5.3 s on average) - approximately the same duration as for the acquisition of the sweep over the Graf plane.

At UMG, recordings were collected from 97 neonates, yielding a total of 582 sweeps, including 194 Graf-standard acquisitions. At UKSH, recordings from 61 neonates were collected according to the same six-sweep protocol. In addition, 50 sweeps were acquired with optical tracking to provide ground-truth probe trajectories for 3D reconstruction. For the tracked acquisitions, we employed an optical tracking system integrated into a dedicated

setup comprising the camera, a processing workstation, and a tracking adapter mounted on the ultrasound probe (reference image of setup in Appendix 12). Tracking data were synchronised with the ultrasound frame stream, and a full calibration was performed for each setup session using a calibration phantom following the method described in (Heyer et al., 2025).

Additional demographic and clinical metadata were collected, including sex, age in days, Graf classifications for both hips, delivery information, and family history. Infants examined at UKSH between 19/06/2025 and 30/10/2025 comprised a cohort of $N = 50$ subjects (16 female, 34 male), with a mean age of 1.6 days (SD 0.78, range 0–4 days).

In the present work, we focus primarily on the tracked subset because the availability of ground-truth probe poses allows quantitative reconstruction evaluation. However, one of the goals of this study is to enable the reconstruction and segmentation of the large untracked UMG dataset using the models and methodology developed here, ultimately extending its utility for downstream research.

The public release of the full dataset is currently under review. In the meantime, we offer to evaluate external reconstruction and segmentation methods on the dataset upon request to facilitate reproducibility and comparison within the community.

## 5.2. 2D Segmentation

### 5.2.1. DATA ANNOTATION

To obtain ground-truth data for training and evaluating the 2D segmentation model, we manually annotated ultrasound frames extracted from 32 freehand sweeps acquired in eight neonates (four sweeps per subject) from the tracked UKSH dataset. Each sweep consisted on average of 57 frames. To reduce annotation workload without substantially compromising anatomical variability, we sampled every third frame, resulting in a total of 611 frames selected for labelling. At this stage all annotations were created entirely by hand and without assistance from automated tools. The labelling process was carried out in close consultation with an experienced paediatric orthopaedic specialist to ensure anatomical correctness and clinical relevance. The selected frames had an original resolution of $727 \times 800$ pixels.

Four structures were delineated as separate classes: the ilium, the femoral head, the chondro-osseous border, and the cortical bone of the proximal femur (Fig. 2). These structures were chosen because they represent echogenic or otherwise clearly identifiable anatomical landmarks that are essential for hip morphology assessment. Bony and cartilaginous interfaces, such as the ilium, cortical bone, and the chondro-osseous border, typically appear with strong acoustic contrast in ultrasound and can therefore be annotated with high reliability. The femoral head, in contrast, is often visualised as a hypoechoic region and may require interpretation rather than strict contour tracing; however, it remains clinically relevant both for Graf-based evaluation and for downstream 3D morphological analysis.

Soft-tissue structures such as tendons and muscles were intentionally excluded from the annotation protocol. These structures are more variable, exhibit lower contrast, and are more time-consuming to label, while offering limited added value for assessing acetabular shape or femoral head coverage.

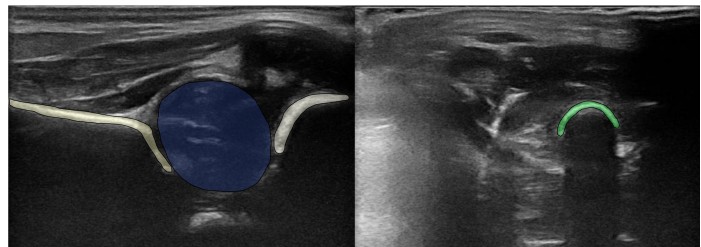

Figure 2: Exemplary manual annotation of posterior-anterior ultrasound sweep over the hip in 2D. Left: ilium (yellow), femoral head (blue), chondro-osseous border (white). Right: cortex of the femur (green).

To ensure annotation quality, the labelled frames underwent a secondary review by an additional rater. This quality check focused on internal consistency of boundary placement and agreement with expected anatomical morphology. Ambiguous or difficult cases were discussed jointly and corrected where necessary. Additionally, we performed a double-blind re-annotation of one exemplary subject comprising four ultrasound sweeps by the two annotators. The case was randomly selected and independently rated by both annotators as comparatively challenging. For this subject, we observed a mean Dice agreement of 0.76 across the four target structures, with the ilium proving the most ambiguous with a Dice of 0.70, likely due to its narrow structure. The agreement was highest for the femoral head with 0.86.

### 5.2.2. MODEL TRAINING

The annotated frames were divided into training, validation, and test sets (432, 108, and 71 images, respectively). This split was kept fixed across all 2D segmentation experiments to ensure comparability between models. For segmentation, we employed the nnU-Net framework, which is widely regarded as the state of the art in medical image segmentation due to its strong performance across modalities and its automatic adaptation of preprocessing, architecture, and training schedules to the dataset at hand. Its robustness to limited data and heterogeneous ultrasound appearance made it particularly suitable for neonatal hip ultrasound. Preliminary tests with zero-shot and interactive segmentation models yielded substantially poorer performance, likely due to the lack of ultrasound data in their pre-training corpora, further supporting nnU-Net as the most reliable approach for this task. We explored three 2D segmentation configurations. The first model (ID 2.1) used the default 2D PlainConvUNet architecture provided by nnU-Net. The second model (ID 2.2) increased the input resolution to better preserve fine anatomical structures while keeping the network architecture identical. Finally, to incorporate temporal and spatial context from neighbouring slices, we implemented a multi-frame 2.5D variant (ID 2.3) in which two preceding and two following frames were included as additional input channels for each prediction. This allowed the model to leverage local sweep context while remaining computationally efficient due to its fully 2D processing pipeline. All models were trained under identical data splits, identical loss formulation, and consistent preprocessing principles de-

fined by nnU-Net, enabling a controlled comparison of how input resolution and temporal context affect segmentation performance. A detailed training description can be found in Appendix E.

### 5.3. 3D Reconstruction

For reconstruction, we employ differentiable voxelisation as described in (Heinrich et al., 2023). Each acquired frame $I_i \in \mathbb{R}^{n_x \times n_y}$ of a sequence $I_0, \ldots, I_n$ is associated with a discrete 2D grid $G_i \in \mathbb{R}^{n_x \times n_y \times 3}$ of 3D coordinates lying in an affine subspace of $\mathbb{R}^3$. To express all grids in a common reference frame, we use affine transformation matrices $T_i \in \mathbb{R}^{4 \times 4}$, $i = 1, \ldots, n$, that encode the relative pose of the probe with respect to a neutral reference position (typically the first frame). As this neutral position is defined solely by acquisition order, it does not convey any anatomical reference. Probe trajectories can be obtained either from sensor-based tracking or image-based freehand reconstruction methods. The grid coordinates $G_i(x, y)$ can then be computed as the affine transformation of identity grid coordinates $(x, y)$ (representing the neutral position) with the affine matrix $T_i$. Based on the coordinates $G_1, \ldots, G_n$, the 2D intensity images $I_0, \ldots, I_n$ are interpolated into the 3D voxel lattice using inverse grid sampling (Heinrich et al., 2023), yielding the reconstructed volume $I_{\text{rec}} \in \mathbb{R}^{n_x \times n_y \times n_z}$. While tracking-based reconstruction provides spatially accurate 3D volumes, its clinical use is limited by the need for external hardware, calibration, and line-of-sight constraints. To enable reconstruction directly from untracked freehand sweeps, we trained a model that infers frame-to-frame transformations from image content, supplemented by auxiliary motion cues when available. This approach aims to generalise reconstruction to the larger untracked dataset and, ultimately, routine clinical acquisitions.

For training, we used the tracked UKSH subset: 177 sweeps from 47 neonates for training and 12 sweeps from three additional subjects for validation. All trajectories were manually inspected and preprocessed to ensure consistent quality. Non-informative sections at the beginning or end of sweeps - caused, for instance, by slow initial probe movement or delayed stopping - were removed, and sweeps with incorrect probe orientation (e.g., unintended 180° flips) were corrected to maintain anatomical consistency. Frame-to-frame motion was predicted using a CNN inspired by Prevost et al. (2018). Optical flow between adjacent frames was included to provide dense displacement cues, and IMU-derived angular velocity stabilised rotation estimates. Training samples consisted of short sequences of six consecutive frames, with transformations predicted in an orthogonalised parameterisation for numerical stability. The final model used an EfficientNet-B0 backbone without ImageNet pretraining and was trained for 200 epochs with a batch size of 32 using an MSE loss on the predicted transformations and AdamW optimisation. The training duration was 5.1 h on a NVIDIA Quadro P6000 GPU (memory usage approximately 12 GB) and the model has 4.069.513 trainable parameters. At inference time, predicted transformations are passed into the differentiable voxelisation pipeline to generate coherent 3D reconstructions. The corresponding 2D segmentation masks from Section 5.2.2 are reconstructed alongside the sweeps to obtain initial 3D segmentation volumes.

### 5.4. 3D Segmentation

To obtain more reliable volumetric training data, 24 reconstructed hip volumes - covering all four acquisition orientations - were manually corrected as an initial step towards 3D segmentation. These refined labels, derived from the first reconstruction stage, were used to train an initial 3D nnU-Net model (ID 3.1). The model followed nnU-Net's fully automated 3D configuration, including its default preprocessing, patch-based training strategy, and loss formulation. To further improve label quality while reducing annotation effort, we adopted a staged refinement procedure. Predictions of model 3.1 were used as the starting point for a second round of manual correction, replacing the less coherent 2D-derived reconstructions. The predicted masks were smoothed prior to annotation to facilitate editing and to reduce the need for voxel-precise correction. This yielded 48 high-quality 3D ground-truth volumes with balanced coverage of acquisition directions, sides, and subjects. These refined labels were used to train a second 3D nnU-Net model (ID 3.2), which operated on the native reconstructed resolution without additional resampling. The dataset was split into training, validation, and test sets (32/8/8 volumes). Model 3.2 again followed nnU-Net's automatically selected 3D architecture and preprocessing pipeline, but using the low-resolution configuration appropriate for the input spacing. Both 3D models were trained with identical optimisation and loss settings, ensuring a fair comparison. Detailed information on the training process is provided in Appendix F.

## 6. Results

### 6.1. Evaluation Metrics

For the quantitative evaluation of the 2D segmentation models, we employed the Dice coefficient (DSC) as the primary performance metric. For the 3D segmentation models, we extended the evaluation to include not only the Dice coefficient but also the Hausdorff distance (HD) and the average surface distance (ASD), providing a more comprehensive assessment of spatial accuracy and boundary alignment in three dimensions. Because full 3D ground truth is inherently limited, all reconstructed segmentation volumes were additionally subjected to a structured qualitative assessment. We used a three-level visual grading scheme to compensate for the lack of exhaustive annotations and to ensure clinical plausibility of the outputs. Segmentations rated as good showed complete and anatomically coherent representations of all major structures (femoral head, ilium, cortical bone, and chondro-osseous transition), with a smooth, approximately spherical or ovoid femoral head and no holes or implausible discontinuities. Volumes rated as adequate still contained all structures but showed local imperfections, partial incompleteness, or mild artefacts. Reconstructions judged as insufficient displayed missing or severely deformed structures, such as only partially visible ilium or a strongly distorted femoral head.

For the reconstruction accuracy itself, we evaluated absolute and relative translational and rotational errors, comparing predicted probe trajectories to the reference tracking data.

### 6.2. 2D Segmentation

To keep computational requirements low, we refrained from multi-fold cross-validation and instead used a fixed data split. All models were trained for 1000 epochs on a NVIDIA

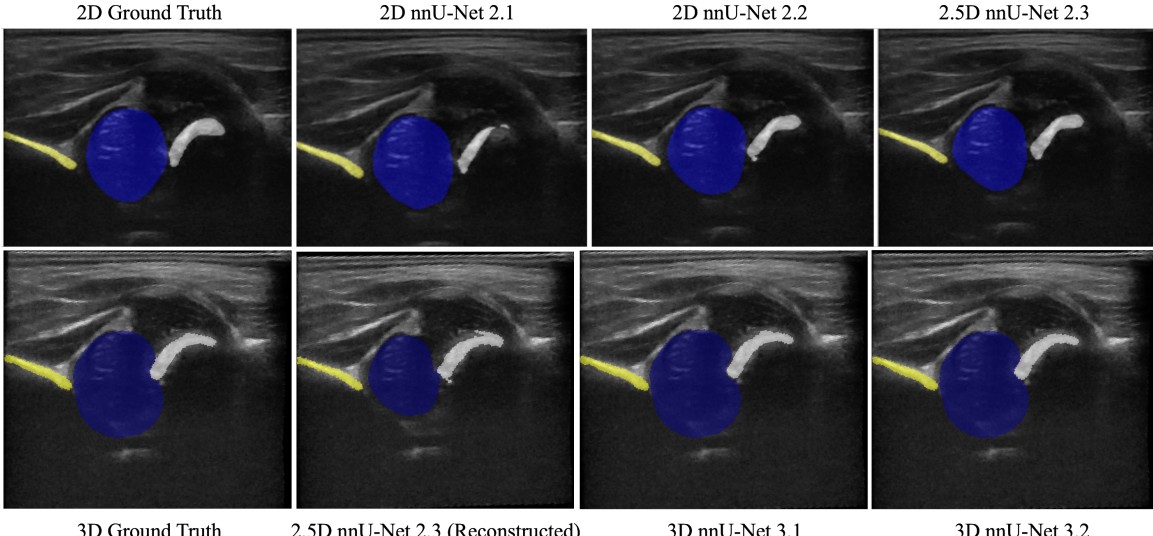

Figure 3: Top: Initial 2D ground truth annotation (left) and segmentation results for the three trained 2D nnU-Net models for ilium (yellow), femoral head (blue) and chondro-osseous border (white). Bottom: Refined 3D ground truth annotation of the approximate same frame as top row, interpolated from 3D reconstruction (left); results of 2D network 2.3 reconstructed and sliced; results of 3D nnU-Nets 3.1 and 3.2.

GeForce RTX 2080 Ti GPU. Performance was evaluated using the Dice coefficient for the four annotated classes (ilium, femoral head, chondro-osseous border, cortical bone) with results summarised in Tab. 1 and Fig. 4. The first configuration (ID 2.1) achieved a mean DSC of 0.631 across all classes and test frames. The second configuration (ID 2.2), which used a larger input resolution and therefore required roughly 23 hours of training, produced a more stable learning curve but a slightly lower mean DSC of 0.588. The multi-frame 2.5D model (ID 2.3) required approximately 30 hours of training and reached the best overall performance with a mean DSC of 0.657. As shown in Fig. 4, all three models achieved consistently high Dice scores for the cortical bone and, to a slightly lesser extent, for the ilium. Performance on the femoral head varied more substantially across configurations, with the second model performing noticeably worse on this class. The chondro-osseous border proved most challenging across all architectures, exhibiting low Dice scores and pronounced variance across test frames.

## 6.3. Reconstruction

We evaluated the performance of the image-based reconstruction model by comparing its predicted inter-frame transformations with the ground-truth trajectories obtained from optical tracking. Qualitative assessment was performed by visualising the predicted probe path alongside the tracked trajectory in 3D space. Fig. 5 illustrates three representative examples, with ground truth shown in green and the model prediction in red. The complexity of the

Table 1: Quantitative evaluation of all trained 2D and 3D segmentation models, reporting the average Dice coefficient across the four target structures.

| Model | DSC | | | |
|---|---|---|---|---|
| | Bone Cortex | Femoral Head | Ilium | CO-Border |
| 2D nnU-Net 2.1 | $0.693_{\pm 0.338}$ | $0.628_{\pm 0.331}$ | $0.707_{\pm 0.263}$ | $0.495_{\pm 0.423}$ |
| 2D nnU-Net 2.2 | $0.682_{\pm 0.341}$ | $0.466_{\pm 0.389}$ | $0.726_{\pm 0.236}$ | $0.479_{\pm 0.428}$ |
| 2.5D nnU-Net 2.3 | $0.671_{\pm 0.354}$ | $0.650_{\pm 0.351}$ | $0.755_{\pm 0.189}$ | $0.554_{\pm 0.406}$ |
| 3D nnU-Net 3.1 | $0.730_{\pm 0.137}$ | $0.608_{\pm 0.353}$ | $0.736_{\pm 0.070}$ | $0.336_{\pm 0.211}$ |
| 3D nnU-Net 3.2 | $\mathbf{0.877_{\pm 0.092}}$ | $\mathbf{0.918_{\pm 0.103}}$ | $\mathbf{0.868_{\pm 0.087}}$ | $\mathbf{0.634_{\pm 0.155}}$ |

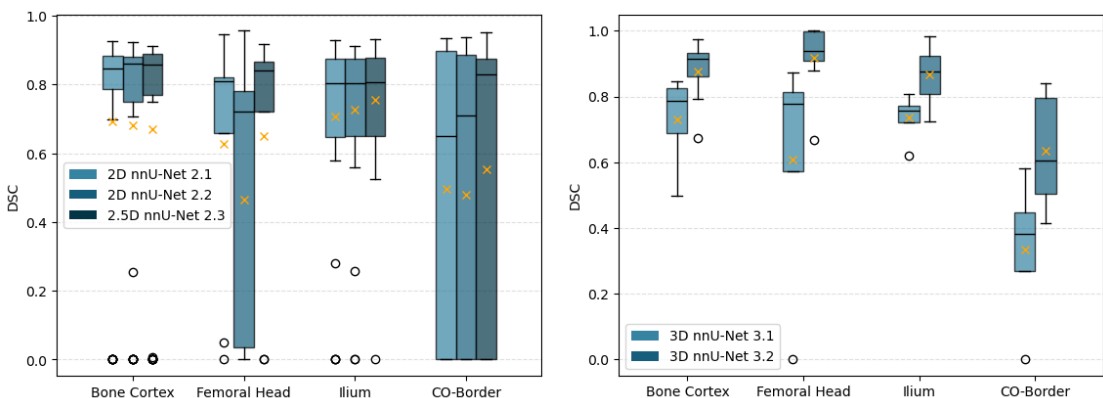

Figure 4: Distribution of Dice coefficients for the four target structures across all 2D and 3D segmentation models, with mean values indicated by orange crosses.

trajectories increases from left to right. The frame-to-frame relative errors of $0.68_{\pm 0.17}\,\mathrm{mm}$ in translation and $0.27^{\circ}_{\pm 0.09}$ in rotation are consistent across all sweep conditions and align with values reported in the literature for learning-based ultrasound pose estimation on other musculoskeletal anatomies. The low rotational errors confirm that IMU-based orientation estimates provide reliable angular information. However, errors accumulate over the sweep trajectory, reaching a final drift of up to $12.60_{\pm 4.96}\,\mathrm{mm}$ in translation. This drift can be attributed to motion artifacts from the infant and inherent challenges in sensorless reconstruction. Analyzing the two sweep directions separately reveals notable differences in final drift. Posterior-to-anterior sweeps (p_a) exhibit lower final drift (up to $8.92_{\pm 2.49}\,\mathrm{mm}$) compared to proximal-to-distal sweeps (p_d) with up to $16.28_{\pm 3.92}\,\mathrm{mm}$. This difference is most pronounced in the lateral (y) direction, where p_d sweeps show substantially higher drift ($13.38_{\pm 2.97}\,\mathrm{mm}$) than p_a sweeps ($2.70_{\pm 1.34}\,\mathrm{mm}$), potentially attributable to infant motion during the longer proximal-to-distal acquisition. An in-depth quantitative evaluation is provided in Appendix A.1.

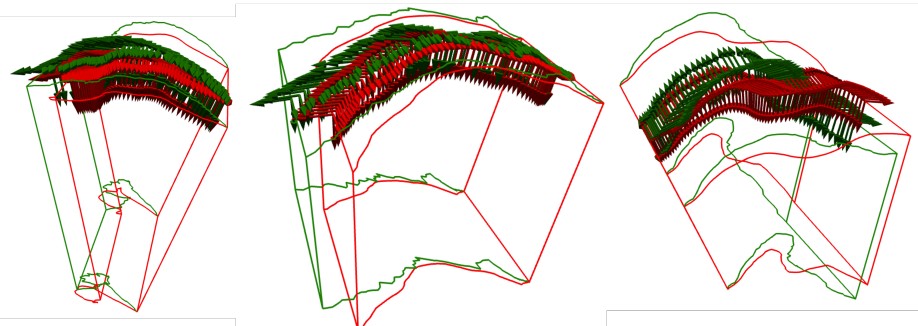

Figure 5: Trajectories of three hip ultrasound sweep examples: ground-truth trajectories obtained via optical tracking (green) and model-predicted trajectories (red).

### 6.4. 3D Segmentation

The reconstruction of the tracked sweeps enabled the generation of initial volumetric segmentation masks using the 2D predictions of model 2.2. An example is shown in Fig. 6. As expected, these initial 3D masks exhibited pronounced slice-wise artefacts – most notably a stepped or streak-like appearance – originating from frame-to-frame inconsistencies in the 2D segmentations. Since the 2D model has no access to spatial context across adjacent frames during inference, such discontinuities propagate into the reconstructed volume. Nevertheless, these coarse 3D masks provide a valuable starting point for manual refinement: instead of annotating entire volumes from scratch, annotators only needed to correct and complete the existing masks. Moreover, this approach yielded pseudo-labels for every frame of a sweep, whereas the original 2D annotations covered only every third frame due to time constraints. Based on these preliminary masks, we trained the first 3D nnU-Net model (ID 3.1). Training for 1000 epochs required approximately 33 hours. Evaluation on four fully annotated test sweeps resulted in a mean Dice score of 0.602 across all classes (Tab. 1). To further improve segmentation quality, all reconstructed volumes were manually refined and smoothed. In addition, the number of available training and test cases was increased by incorporating more corrected volumes. Using these improved annotations, we trained a second 3D nnU-Net (ID 3.2), again for 1000 epochs, with a total training time of approximately 42 hours. This model achieved a substantially higher mean Dice score of 0.824 on eight test volumes, with consistently improved performance across all anatomical classes (Tab. 1). Beyond volumetric Dice, we assessed boundary accuracy using HD and ASD. The second 3D model achieved a mean HD of 1.67 mm, and mean ASD of 0.31 mm (ground-truth to prediction) and 0.24 mm (prediction to ground-truth), indicating a close geometric correspondence between predicted and reference shapes. Finally, to assess model robustness on unlabelled sweeps, we conducted a structured visual evaluation as described in Section 6.1. Across 74 sweeps, 53 % were rated good, 39 % adequate, and 8 % insufficient. Typical failures included incomplete representation of osseous structures and motion-related artefacts arising from irregular probe movement. The final 3D segmentation model weights (ID 3.2) are publicly available for inference at https://github.com/MDL-UzL/JOINT.

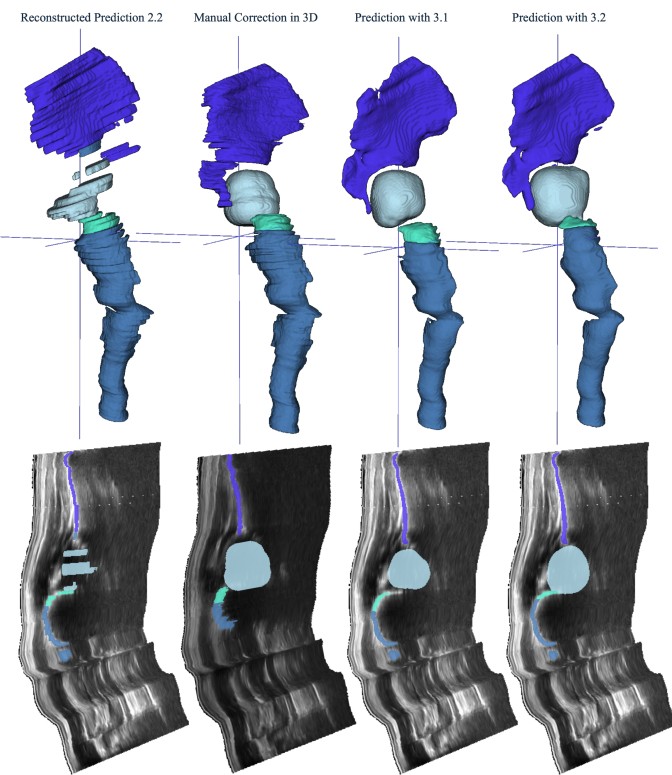

Figure 6: Progression of segmentation performance from left to right: reconstructed 2D predictions, manual corrections, predictions from the first-stage 3D model, and final predictions from the second-stage 3D model. **Top:** volume renderings of the four target structures: ilium (purple), femoral head (grey), chondro-osseous border (turquoise), and bone cortex (blue). **Bottom:** representative slices through reconstructed sweeps with segmentation masks.

## 7. Discussion

Our study presents one of the first systematic evaluations of combined 2D and 3D deep-learning pipelines for neonatal hip ultrasound in the context of developmental dysplasia of the hip. While most prior work focuses on single-frame 2D segmentation, our approach integrates volumetric acquisition, image-based reconstruction and 3D segmentation on a multi-centre clinical dataset, thereby addressing important gaps in current research.

**Clinical and dataset contributions**: A major strength of this work is the creation of a real-world neonatal hip dataset from two hospitals, acquired with a consistent protocol and including Graf-standard planes as well as orthogonal freehand sweeps. The availability of optical tracking for a subset of sweeps provides ground truth for evaluating reconstruction quality and offers a more clinically relevant benchmark than existing volunteer datasets such as TUSRec (Li et al., 2025). The demographic and clinical metadata further support future studies on developmental variability and risk modelling.

**Annotation quality**: Manual annotation of 2D ultrasound data is inherently challenging due to speckle noise, acoustic shadowing, limited contrast, and strong viewpoint dependency. As a result, purely slice-wise 2D annotations are prone to inconsistencies and exhibit higher inter-rater variability compared to annotations performed in a spatially coherent 3D context. This limitation is reflected in the moderate inter-rater agreement observed for the initial 2D annotations.

Importantly, our work demonstrates that high-quality 3D annotations emerge not from manual labeling alone, but from the combination of expert annotations and model-based inference followed by iterative refinement in 3D. The integration of automatic predictions with anatomical context across sweeps allows for the correction of slice-wise ambiguities and substantially improves annotation consistency. Consequently, the final 3D annotations reach an inter-rater agreement level comparable to the achieved model performance, indicating that the model operates within the bounds of human annotation variability.

**Segmentation performance**: The 2D experiments highlight the limitations of purely frame-based segmentation for hip ultrasound. While nnU-Net performed well on high-contrast structures such as the ilium and cortical bone, performance dropped notably for low-contrast structures such as the chondro-osseous border and femoral head. The 2.5D multi-frame variant improved performance, underscoring the value of contextual information in resolving ambiguities caused by speckle, shadowing, and probe motion – an observation consistent with previous ultrasound literature.

**Improving consistency in 3D refinements**: 2D predictions led to considerable inconsistencies when reconstructed into 3D volumes, manifesting as discontinuities and surface artefacts. Our staged refinement strategy, using 3D reconstructions as a basis for manual correction and subsequent 3D training, proved effective. Even with a modest number of refined volumes, the final 3D nnU-Net achieved substantially higher Dice scores, exceeding 0.85 for most structures. This confirms that volumetric context is crucial for anatomically coherent hip segmentation.

**Impact of reconstruction drift on clinical measurements**: While we report a mean global translational drift of approximately 12 mm over entire ultrasound sweeps, this metric should be interpreted with caution in the clinical context. The drift reflects an accumulated reconstruction error across the full acquisition trajectory and does not directly translate to local inaccuracies at the hip joint. In particular, we observe notable differences between sweep orientations: posterior–anterior sweeps, although longer in duration, cover a shorter physical distance and provide denser image information, resulting in lower drift compared to proximal–distal sweeps that traverse a larger anatomical range more rapidly.

Crucially, the clinically relevant region for hip assessment is spatially confined to the vicinity of the femoral head and acetabulum. Within this region, visual inspection indicates substantially higher local consistency than suggested by the global drift metric. As Graf angle measurements depend on local anatomical geometry rather than the absolute global pose of the reconstructed volume, moderate global drift does not necessarily imply a proportional impact on clinical measurements.

Finally, infant motion during scanning represents an additional confounding factor. Movements such as leg flexion affect the optical tracking reference itself, such that part of the observed drift may reflect discrepancies between probe tracking and anatomical motion rather than reconstruction errors. In these scenarios, image-based reconstruction may

partially compensate for tracking inconsistencies despite exhibiting higher numerical drift. A dedicated quantitative analysis of the relationship between reconstruction drift and downstream clinical angle measurements therefore constitutes an important direction for future work.

**Translational relevance and robustness**: Although our methods rely on established components rather than novel architectures, we demonstrate a practically relevant workflow that integrates multi-centre acquisition, tracked and untracked sweeps, detailed preprocessing and reproducible training pipelines. Because data were acquired during routine screening, the approach is well aligned with clinical practice. The resulting system provides interpretable outputs suitable for downstream diagnostic tools and offers a foundation for future 3D morphological analyses.

**Limitations and future work**: While our study establishes a robust pipeline for 3D reconstruction and segmentation of neonatal hips, several limitations remain. Manual annotations on 2D ultrasound are inherently challenging and prone to variability, and reconstruction drift over multi-sweep acquisitions can introduce global misalignments, even if local anatomical consistency is higher. Additionally, motion of the infant during scanning adds further uncertainty to both tracking and image-based reconstructions.

Despite these challenges, combining iterative model-based refinement with manual annotation enables the creation of high-quality 3D labels that are substantially more reliable than initial 2D annotations alone. Building on this foundation, future work will focus on evaluating downstream clinical metrics, such as 3D-derived Graf angles, and systematically quantifying the impact of residual reconstruction errors and annotation variability on clinical decision-making.

Our ongoing efforts focus on refining reconstruction, improving multi-sweep fusion, and developing registration strategies to combine orthogonal sweeps into a coherent 3D hip model. This study lays the groundwork for more robust volumetric analysis and, ultimately, more reliable automated DDH assessment.

## Acknowledgments

This work was supported by the German Federal Ministry of Research, Technology, and Space (BMFTR) under grant number 16SV9254.

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

# Appendix A. Performance Metrics

## A.1. Reconstruction

Table 2: 3D ultrasound reconstruction performance. Translational errors in mm, rotational errors in degrees. Values reported as mean$_{\text{std}}$.

| Metric | Translation [mm] | | | | Rotation [°] |
|---|---|---|---|---|---|
| | Total | x | y | z | Total |
| Relative | $0.68_{\pm 0.17}$ | $0.21_{\pm 0.06}$ | $0.34_{\pm 0.12}$ | $0.44_{\pm 0.13}$ | $0.27_{\pm 0.09}$ |
| Absolute (mean) | $5.16_{\pm 1.97}$ | $2.26_{\pm 1.23}$ | $2.80_{\pm 2.08}$ | $2.40_{\pm 1.33}$ | $1.25_{\pm 0.41}$ |
| Final drift | $12.60_{\pm 4.96}$ | $6.46_{\pm 3.58}$ | $8.04_{\pm 5.99}$ | $6.87_{\pm 2.60}$ | $2.60_{\pm 0.61}$ |

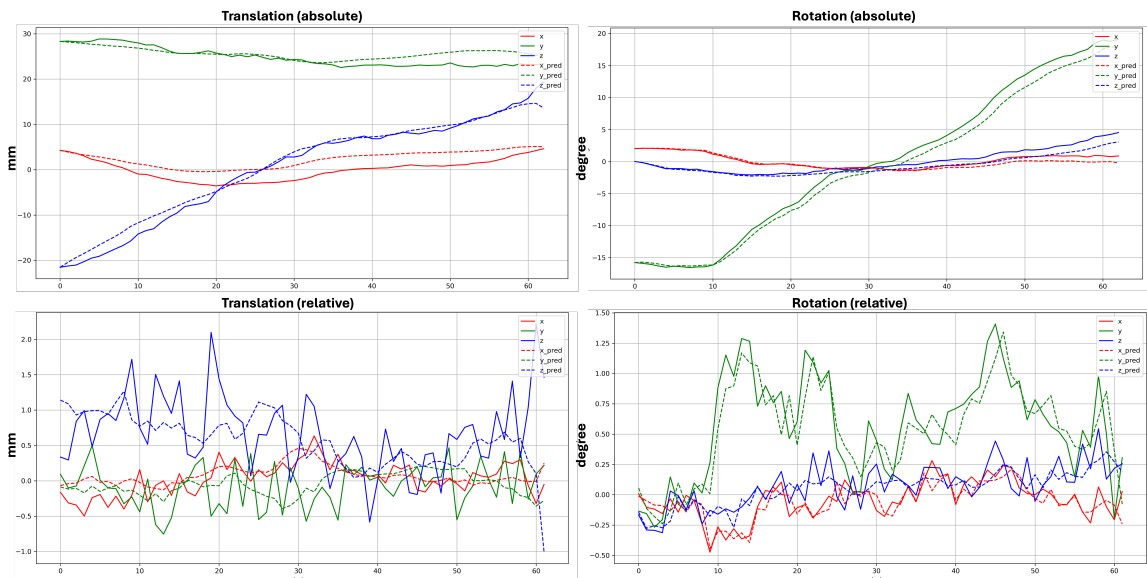

Figure 7: Per-frame reconstruction metrics for the **left** sweep (cf. Fig. 5). Translation error (mm) and rotation error (deg) are shown per ultrasound frame.

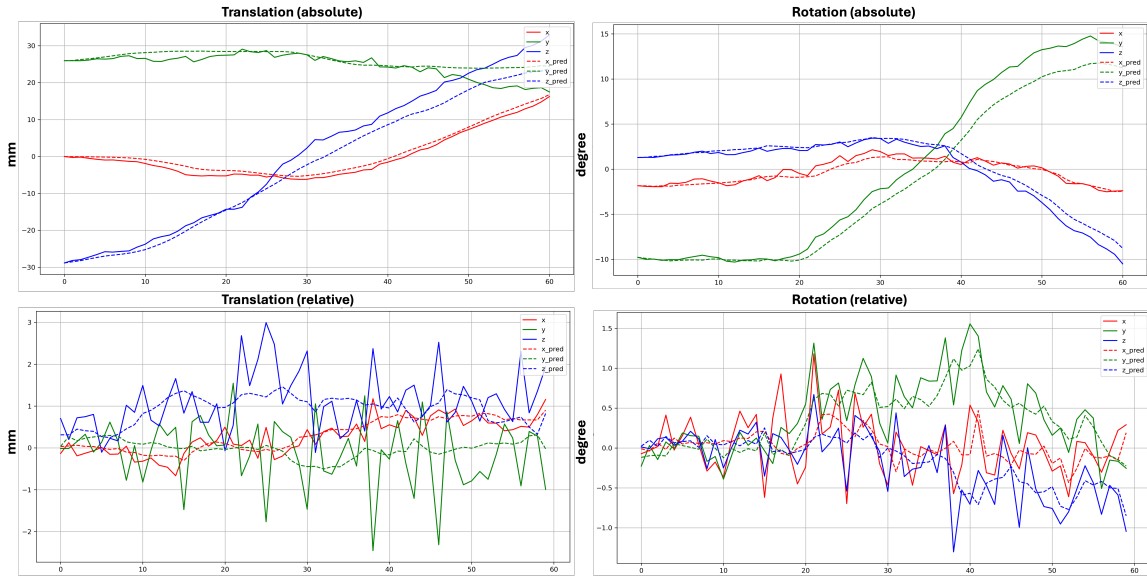

Figure 8: Per-frame reconstruction metrics for the **middle** sweep (cf. Fig. 5). Translation error (mm) and rotation error (deg) are shown per ultrasound frame.

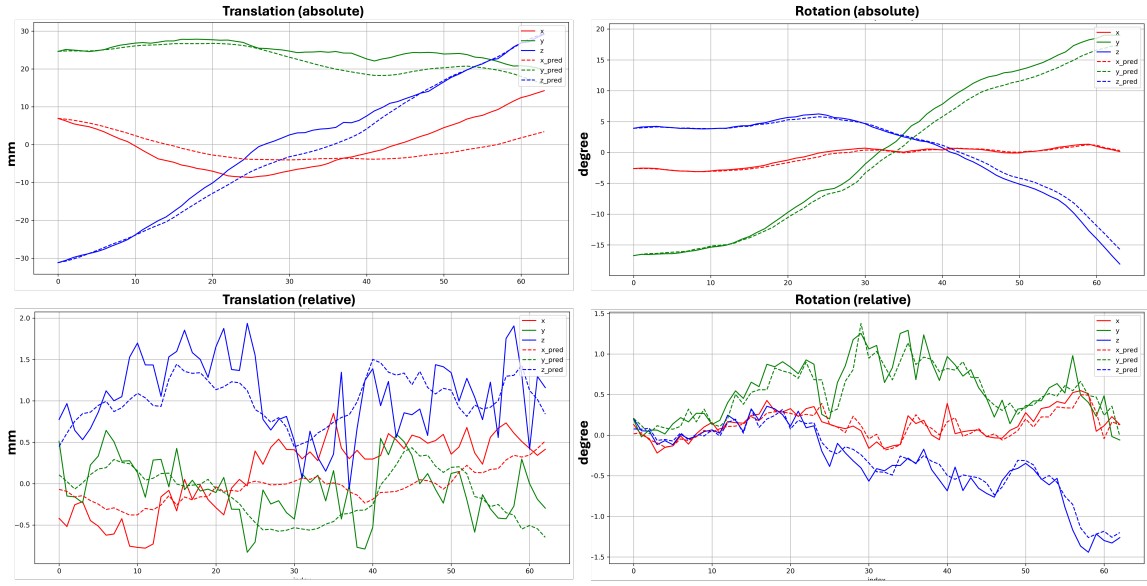

Figure 9: Per-frame reconstruction metrics for the **right** sweep (cf. Fig. 5). Translation error (mm) and rotation error (deg) are shown per ultrasound frame.

## A.2. Segmentation

Table 3: Segmentation performance of the second-stage 3D nnU-Net (ID 3.2) for all target structures, separated by sweep orientation. Metrics include Dice coefficient, Hausdorff distance (mm), and average surface distance (mm).

| Sweep Orientation | Bone Cortex | Femoral Head | Ilium | CO-Border |
|---|---|---|---|---|
| **Dice Coefficient** | | | | |
| Proximal-Distal | 0.938 | 0.873 | 0.869 | 0.614 |
| Posterior-Anterior | 0.817 | 0.963 | 0.866 | 0.653 |
| **Hausdorff-Distance (mm)** | | | | |
| Proximal-Distal | 0.248 | 2.186 | 2.488 | 1.489 |
| Posterior-Anterior | 2.191 | 0.534 | 1.572 | 2.626 |
| **Average Surface Distance (mm)** | | | | |
| Proximal-Distal | 0.037 | 0.686 | 0.359 | 0.372 |
| Posterior-Anterior | 0.209 | 0.165 | 0.088 | 0.564 |

Table 4: Detailed segmentation results of model ID 3.2: Dice coefficients, Hausdorff distances, and average surface distances (ASD) for each test case (rows). The last row shows the mean values for each class across all cases.

| Bone Cortex | Femoral Head | Ilium | CO-Border |
|:---:|:---:|:---:|:---:|
| **DSC** | | | |
| 0.9361 | 0.9446 | 0.9053 | 0.6401 |
| 0.7933 | 0.9330 | 0.8902 | 0.5694 |
| 0.8847 | 0.9992 | 0.9835 | 0.7884 |
| 0.9114 | 0.8800 | 0.8647 | 0.4807 |
| 0.6729 | 0.9190 | 0.8184 | 0.4145 |
| 0.9298 | 0.6679 | 0.7235 | 0.5120 |
| 0.9753 | 0.9995 | 0.9830 | 0.8232 |
| 0.9160 | 0.9992 | 0.7731 | 0.8404 |
| **0.8774** | **0.9178** | **0.8677** | **0.6336** |
| **Hausdorff Distance** | | | |
| 0.2483 | 0.5552 | 0.4966 | 0.8953 |
| 2.6627 | 1.0238 | 1.3371 | 3.7983 |
| 1.0534 | 0.0000 | 0.0000 | 1.2415 |
| 0.4966 | 2.4706 | 1.0238 | 2.2621 |
| 3.4584 | 1.1104 | 1.7905 | 3.9338 |
| 0.2483 | 5.7163 | 8.4312 | 1.7732 |
| 0.0000 | 0.0000 | 0.0000 | 1.0238 |
| 1.5899 | 0.0000 | 3.1603 | 1.5306 |
| **1.2197** | **1.3595** | **2.0300** | **2.0573** |
| **ASD (gt→pred / pred→gt)** | | | |
| 0.0322 / 0.0651 | 0.2199 / 0.2221 | 0.1162 / 0.0773 | 0.3630 / 0.2256 |
| 0.3437 / 0.3060 | 0.2729 / 0.3016 | 0.0695 / 0.2189 | 0.9236 / 0.5603 |
| 0.1636 / 0.0257 | 0.0018 / 0.0014 | 0.0048 / 0.0098 | 0.0509 / 0.2812 |
| 0.0401 / 0.0902 | 0.6345 / 0.4781 | 0.1958 / 0.1256 | 0.6826 / 0.3355 |
| 0.1326 / 0.7950 | 0.3854 / 0.3535 | 0.1298 / 0.3265 | 1.2364 / 0.0926 |
| 0.0505 / 0.0333 | 1.8874 / 1.0631 | 1.0975 / 0.1036 | 0.3780 / 0.4719 |
| 0.0268 / 0.0110 | 0.0005 / 0.0009 | 0.0277 / 0.0180 | 0.0630 / 0.2130 |
| 0.1972 / 0.0215 | 0.0007 / 0.0009 | 0.1491 / 0.5047 | 0.0441 / 0.2547 |
| **0.1233 / 0.1685** | **0.4254 / 0.3027** | **0.2238 / 0.1730** | **0.4677 / 0.3043** |

## Appendix B. Qualitative Segmentation Results

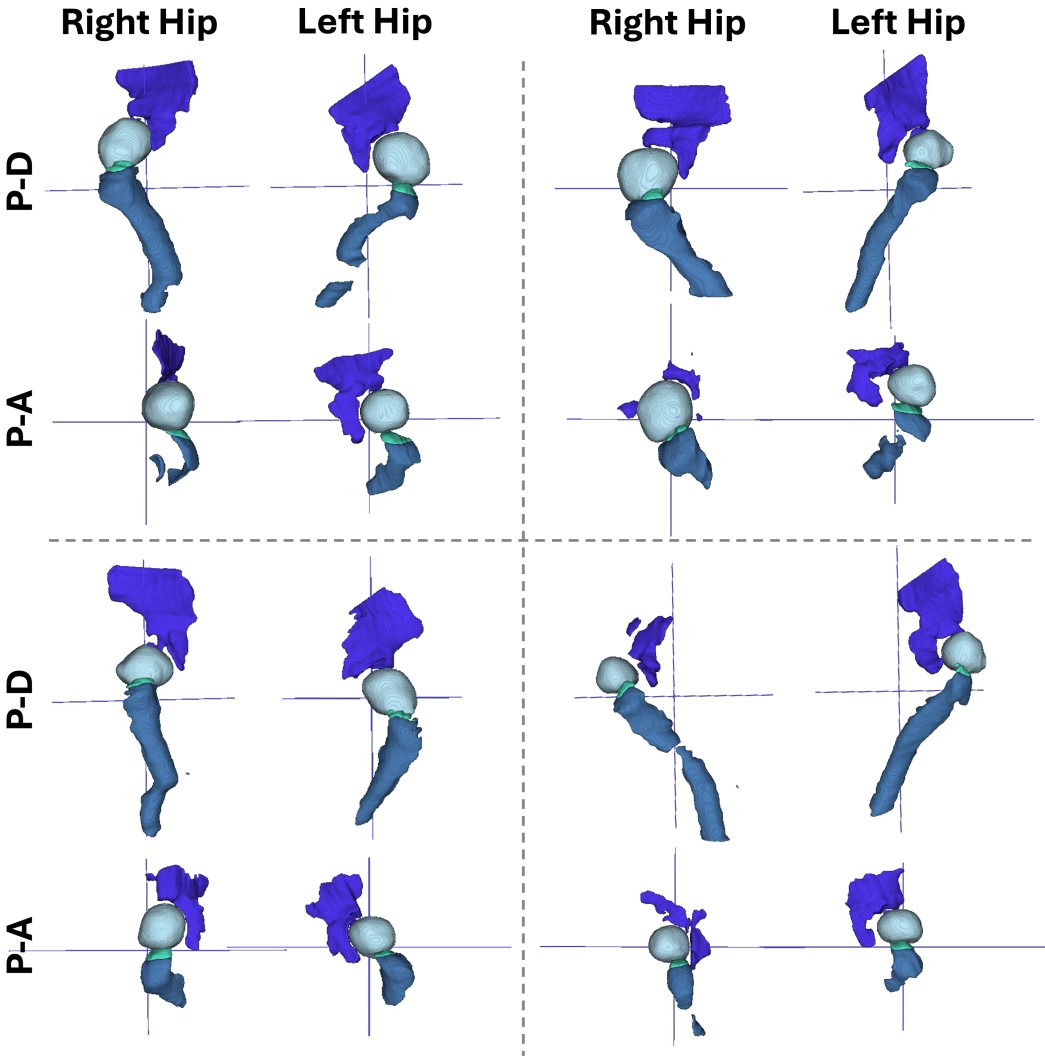

Figure 10: Detailed 3D segmentation examples for 4 unique cases. Displayed are proximal-distal sweeps (P-D) and posterior-anterior sweeps (P-A) over the left and right hip. Segmentations are created with model ID 3.2.

## Appendix C. Standardised 3D Evaluation of 2D and 3D Segmentation Models

To enable a more standardised comparison between 2D and 3D segmentation approaches, we performed an additional evaluation in which all trained 2D nnU-Net models were applied frame-wise to the final set of 3D-corrected test sweeps. The resulting 2D segmentations were subsequently reconstructed into 3D volumes using the same reconstruction pipeline as for the 3D models and evaluated volumetrically against the refined 3D ground truth annotations.

This evaluation mitigates the domain mismatch between slice-wise 2D metrics and volumetric 3D Dice scores reported in the main manuscript (Tab. 1) and allows a fairer comparison under a unified 3D evaluation protocol. Quantitative results of this standardised 3D assessment are summarised in Tab. 5. We note that computing 2D Dice scores on interpolated slices from 3D volumes was deliberately avoided, as freehand ultrasound trajectories do not provide a one-to-one correspondence between acquisition frames and tomographic planes. Such an evaluation would therefore have limited interpretability.

Table 5: Quantitative evaluation of all trained 2D and 3D segmentation models, reporting the average Dice coefficient across the four target structures. The last column shows the mean Dice across all labels including standard deviation.

| Model | Bone Cortex | Femoral Head | Ilium | CO-Border | Mean DSC |
|---|---|---|---|---|---|
| 2D nnU-Net 2.1 | $0.488_{\pm 0.339}$ | $0.669_{\pm 0.115}$ | $0.592_{\pm 0.127}$ | $0.255_{\pm 0.118}$ | $0.501_{\pm 0.101}$ |
| 2D nnU-Net 2.2 | $0.488_{\pm 0.339}$ | $0.669_{\pm 0.116}$ | $0.592_{\pm 0.127}$ | $0.255_{\pm 0.118}$ | $0.501_{\pm 0.101}$ |
| 2.5D nnU-Net 2.3 | $0.521_{\pm 0.297}$ | $0.649_{\pm 0.152}$ | $0.595_{\pm 0.119}$ | $0.264_{\pm 0.164}$ | $0.507_{\pm 0.108}$ |
| 3D nnU-Net 3.1 | $0.650_{\pm 0.311}$ | $0.819_{\pm 0.262}$ | $0.718_{\pm 0.156}$ | $0.399_{\pm 0.143}$ | $0.646_{\pm 0.149}$ |
| 3D nnU-Net 3.2 | $\mathbf{0.863_{\pm 0.090}}$ | $\mathbf{0.906_{\pm 0.105}}$ | $\mathbf{0.851_{\pm 0.081}}$ | $\mathbf{0.607_{\pm 0.147}}$ | $\mathbf{0.807_{\pm 0.111}}$ |

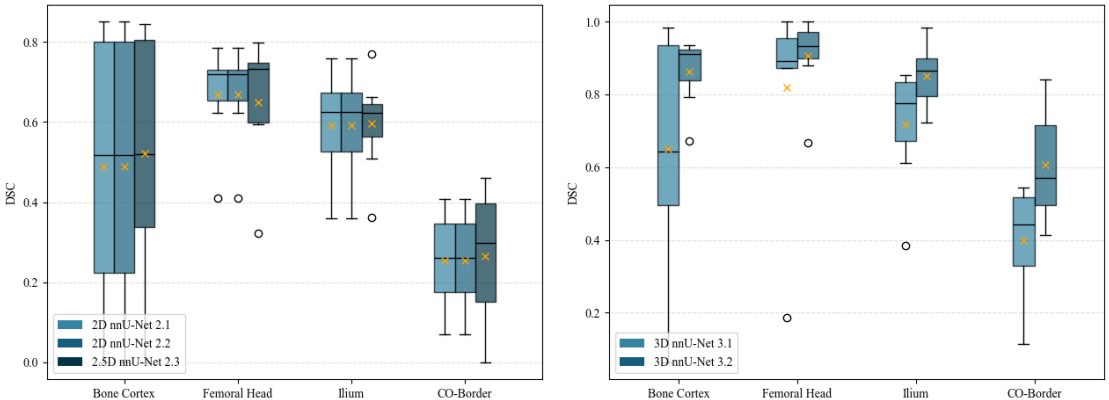

Figure 11: Distribution of Dice coefficients for the four target structures across all 2D and 3D segmentation models, with mean values indicated by orange crosses.

## Appendix D. Dataset & Acquisition Characteristics

Table 6: Characteristics of the hip ultrasound dataset with patients examined between 19/06/2025 and 30/10/2025 at the Universitätsklinikum Schleswig-Holstein, Campus Kiel. ID, sex, age in days, Graf type left and right, family history, and delivery information given in the table for $N = 50$ patients. Age statistics: Mean: 1.6 days, Std: 0.78 days, Min: 0 days, Max: 4 days. $N_{female} = 16$, $N_{male} = 34$

| ID | Sex | Age (d) | Graf type right | Graf type left | Family history | Delivery |
|---|---|---|---|---|---|---|
| J3GRLN | f | 3 | Ib | Ib | Hip dysplasia father's sister | C-section |
| PBMV76 | m | 2 | Ib | IIa | Pelvic tilt father | C-section |
| BO96VQ | f | 1 | Ib | Ib | | normal |
| UWMKHQ | m | 1 | Ib | Ib | | C-section |
| 823EAS | m | 2 | Ib | Ib | | normal |
| KW65JF | m | 1 | Ib | Ib | | normal |
| 1WY90P | f | 2 | IIa | IIa | | normal |
| XBVCQQ | m | 1 | Ib | Ib | | C-section |
| J4QAMM | m | 1 | Ib | Ib | | normal |
| GIC583 | m | 2 | Ib | Ib | | normal |
| FKMJ7C | m | 2 | IIa | IIa | | C-section |
| ETR98D | f | 2 | Ib | Ib | | normal |
| CT8QLI | f | 2 | Ib | Ib | | normal |
| O460DY | m | 2 | Ib | Ib | | normal |
| Q1A5KG | m | 1 | Ib | Ib | | normal |
| XO4B0A | f | 1 | Ib | Ib | | C-section |
| B8B9AF | m | 1 | Ib | Ib | | normal |
| U78BLH | m | 2 | Ib | Ib | | normal |
| 7V4A6U | f | 2 | IIa | IIa | | normal |
| RH2PZ9 | f | 2 | Ib | Ib | | normal |
| 71JZK7 | m | 1 | Ib | IIa | | C-section |
| 7S78PO | f | 1 | IIa | IIa | | normal |
| 0Y3T5K | f | 2 | Ib | Ib | | normal |
| 8Q71TT | m | 1 | Ib | Ib | | normal |
| 3GL2MU | m | 1 | Ib | Ib | | C-section |
| WF8976 | f | 2 | Ib | Ib | | normal |
| JILV0C | m | 3 | Ib | Ib | | C-section |
| BCODO6 | m | 2 | Ib | Ib | | normal |
| MMU98Z | m | 1 | Ib | Ib | | normal |
| KVN8FZ | m | 2 | Ib | Ib | | normal |
| OS2ENE | f | 3 | Ib | Ib | | normal |
| LKPC9Z | m | 2 | Ib | Ib | | normal |
| TVU7J9 | f | 2 | Ib | Ib | | normal |
| 17FOUD | m | 2 | Ib | Ib | | normal |
| CB0ZUM | f | 1 | Ib | Ib | | normal |
| C2WIY1 | m | 1 | Ib | Ib | | C-section |
| M52CPP | m | 1 | Ib | Ib | | normal |
| 50EXN9 | m | 1 | IIa | Ib | | C-section |
| 8S4NGK | m | 2 | Ib | Ib | | C-section |
| J3VLDM | m | 1 | Ib | Ib | | C-section |
| IMCNNZ | m | 1 | Ib | Ib | | C-section |
| EB3WZL | m | 4 | Ib | Ib | | normal |
| 0IRBPU | m | 3 | Ib | Ib | | C-section |
| 4YU2OP | f | 1 | Ib | Ib | | normal |
| U7QL3N | m | 1 | Ib | Ib | | normal |
| Y9N6HY | m | 1 | Ib | Ib | | normal |
| ZOX664 | m | 0 | IIa | IIa | | normal |
| VGLXJL | m | 0 | Ib | Ib | | normal |
| W43ABL | m | 2 | Ib | Ib | | normal |
| ALC2IT | f | 2 | Ib | Ib | | normal |

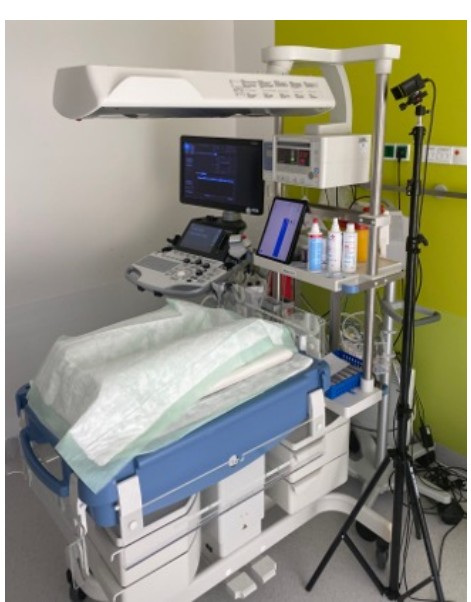

Figure 12: Data Acquisition Setup. The child is placed into a positioning cradle to minimise movement during scanning. Ultrasound probe movement can be tracked using a tracking camera (right) while imaging results are displayed in a screen above the cradle.

## Appendix E. Detailed Training Parameters for 2D Segmentation Models

### E.1. Model ID 2.1 – Baseline 2D nnU-Net

- **Architecture:** 7-stage PlainConvUNet (default nnU-Net 2D)

- **Convolution blocks:** 2 per stage; Leaky ReLU activation; instance normalisation

- **Input resolution:** $256 \times 256$

- **Batch size:** 27

- **Preprocessing:**

    - Z-score normalisation (mean $\approx 131$, SD $\approx 57$)
    - Bicubic interpolation for images (order $= 3$)
    - Nearest-neighbour interpolation for masks (order $= 1$)

- **Training:**

    - 1,000 epochs
    - Optimiser: Stochastic Gradient Descent
    - Loss: Soft Dice + Cross-Entropy
    - Training duration: 10 h 14 min
    - Number of trainable parameters: 92.468.062
    - Memory less than 6 GB on NVIDIA GeForce RTX 2080 Ti

### E.2. Model ID 2.2 – High-Resolution 2D nnU-Net

- **Architecture:** As in Model 2.1

- **Feature widths:** 32, 64, 128, 256, 512, 512, 512

- **Input resolution:** $384 \times 448$ (median: $363 \times 400$)

- **Batch size:** 19

- **Preprocessing:**

    - Z-score normalisation (mean $\approx 132$, SD $\approx 57$)

- **Training differences:**

    - Nesterov momentum enabled
    - `batch_dice` = True (to emphasise small structures)
    - Training duration: 23 h 15 min
    - Number of trainable parameters: 92.468.062
    - Memory approx. 6 GB on NVIDIA GeForce RTX 2080 Ti

### E.3. Model ID 2.3 – Multi-Frame 2.5D nnU-Net

- **Input:** 5 consecutive frames provided as 5-channel 2D input

- **Per-channel preprocessing:** Independent Z-score normalisation (mean $\approx$ 106, SD $\approx$ 57 for each channel)

- **Architecture:** Identical to Model 2.2

- **Input patch size:** $384 \times 448$ (median: $363 \times 400$)

- **Training and preprocessing:**

  - All remaining parameters identical to Model 2.2
  - Training duration: 29 h 40 min
  - Number of trainable parameters: 92.468.062
  - Memory approx. 6 GB on NVIDIA GeForce RTX 2080 Ti

## Appendix F. Detailed Training Parameters for 3D Segmentation Models

### F.1. Model ID 3.1 – First-Stage 3D nnU-Net

- **Input volumes:** 24 manually corrected 3D reconstructions

- **Resampling:** All volumes resized to $256 \times 332 \times$ depth voxels

- **nnU-Net configuration:** `3d_fullres`

- **Trainer:** `nnUNetTrainer_noDummy2DDA`

- **Architecture:**

  - 6 convolutional stages
  - $3 \times 3 \times 3$ kernels
  - Leaky ReLU activations

- **Patch size:** $112 \times 160 \times 128$ voxels

- **Batch size:** 2

- **Preprocessing:**

  - DefaultPreprocessor (nnU-Net)
  - Z-score normalisation (mean $\approx 69$, SD $\approx 57$)
  - Bicubic interpolation for volumes

- **Training:**

  - 1,000 epochs
  - Same optimiser and loss as 2D models (Soft Dice + Cross-Entropy; nnU-Net defaults)
  - Training duration: 33 h 14 min
  - Number of trainable parameters: 88.213.209
  - Memory approx. 12 GB on NVIDIA GeForce RTX 2080 Ti

### F.2. Second Annotation Stage – Improved Ground-Truth

- **Starting point:** Predictions from Model 3.1

- **Smoothing:** Gaussian filtering prior to manual correction

- **Final dataset:** 48 manually refined 3D segmentations

- **Dataset balance:** Equal distribution of acquisition directions, sides, and subjects

**F.3. Model ID 3.2 – Second-Stage 3D nnU-Net**

- **Input volumes:** 48 refined 3D segmentations

- **No resampling:** Native resolution used

- **Dataset split:** 32 training, 8 validation, 8 test volumes

- **Input statistics:**

    - Median resolution: $202 \times 202 \times 197$ voxels
    - Median isotropic spacing: $0.2483 \times 0.2483 \times 0.2483$ mm$^3$

- **nnU-Net configuration:** `3d_lowres`

- **Architecture:**

    - 6 stages with feature depths: 32, 64, 128, 256, 320, 320
    - $3 \times 3 \times 3$ kernels
    - Leaky ReLU activations

- **Patch size:** $128 \times 128 \times 128$ voxels

- **Batch size:** 2

- **Preprocessing:**

    - Z-score normalisation (mean $\approx 69$, SD $\approx 58$)
    - Bicubic interpolation (images), nearest-neighbour (masks)

- **Training:**

    - 1,000 epochs
    - Identical loss and optimisation settings to Model 3.1
    - Training duration: 42 h 10 min
    - Number of trainable parameters: 88.622.809
    - Memory approx. 12 GB on NVIDIA GeForce RTX 2080 Ti

- **Access:** To support reproducibility and downstream research, the final 3D segmentation model weights are available at https://github.com/MDL-UzL/JOINT.

