# OpenReview forum: "Evaluation of 3D Ultrasound Reconstruction and 2D/3D Segmentation for Neonatal Hip Dysplasia Screening"
_MIDL.io/2026/Validation_Papers — MIDL 2026 - Validation Papers Poster_

### Official Review · Reviewer_V3Pq · 2026-01-01

**Confidence:** 3
**Preliminary Rating:** 4

**Summary:**

This paper presents a clinically oriented validation of a workflow for 3D reconstruction and segmentation of neonatal hips using standard 2D ultrasound probes. The authors acquired a dataset of 50 optically tracked and 150 untracked freehand sweeps. They evaluated the performance of 2D, 2.5D, and 3D nnU-Net architectures for segmenting key hip structures. The study introduces a "staged refinement" strategy where initial 2D segmentations are reconstructed, manually corrected, and used to train a robust 3D segmentation model. Results indicate that volumetric segmentation significantly outperforms slice-based 2D methods, particularly for the femoral head and chondroosseous border.

**Strengths:**

+ The acquisition of 50 optically tracked sweeps to serve as ground truth for probe trajectories is a significant contribution.
+ The validation goes beyond simple Dice scores. The authors include Hausdorff Distance (HD) and Average Surface Distance (ASD) for 3D analysis, and they rigorously evaluate reconstruction error in terms of translational and rotational drift.
+ The study addresses a clear clinical need (limitations of 2D Graf method) and utilizes real-world clinical data from two centers.

**Weaknesses:**

- The initial 2D segmentation model was trained and evaluated on sweeps from only eight neonates14. For a paper positioning itself as a validation study, relying on N=8 subjects for the foundational 2D component is a limitation regarding generalization to anatomical variability across a wider population.
- The reported final translational drift is $12.60 \pm 4.96$ mm. Given that the neonatal hip is a small anatomical structure, a drift >1cm is significant. While the authors acknowledge this challenge, the paper lacks a discussion on whether this margin of error renders the 3D reconstruction unusable for precise angle measurements (Graf angles) which define the diagnosis.
- The paper mentions that annotations underwent a secondary review by an additional rater, but it does not quantify inter-rater variability (e.g., Dice agreement between annotators).
- The motivation for the work is improving DDH screening. However, the validation stops at geometric segmentation accuracy. There is no evaluation of derived clinical metrics (e.g., calculating the Alpha/Beta angles on the 3D models or detecting dysplasia cases vs. controls).

**Detailed Comments:**

See weakness.

**Justification Of The Preliminary Rating:**

The paper is a solid validation study that introduces a high-value dataset and a pragmatic workflow for 3D hip analysis. The use of ground-truth tracking for validating reconstruction is a significant strength that sets this work apart from many "tracker-less" papers that rely solely on reprojection error. However, the validation is somehow limited by the small subject count for the initial segmentation training (N=8) and the lack of "downstream" clinical validation (i.e., proving that the 3D model actually aids in diagnosing DDH via angle measurement, rather than just segmenting pixels). Addressing the inter-rater variability and the clinical implications of the reconstruction drift in the rebuttal would strengthen the paper significantly.

**Questions To Address In The Rebuttal:**

- Can you provide quantitative metrics (e.g., Dice or surface distance) regarding the agreement between the primary annotator and the secondary reviewer? This is crucial to contextualize the reported model Dice scores of ~0.87.
- With a mean drift of ~12mm, how does this deformation affect the reliability of clinical measurements?
- The 2D model used 8 patients. Did you observe any performance degradation when applying this model to the broader UMG cohort (97 neonates)  during the reconstruction phase?

---

### Official Review · Reviewer_PD55 · 2026-01-09

**Confidence:** 4
**Preliminary Rating:** 4
**Final Rating:** 5

**Summary:**

The authors present a unified framework that combines 3D ultrasound reconstruction with 2D/3D nnU-Net–based segmentation for assessing developmental dysplasia of the hip (DDH) in infants. By moving beyond conventional 2D slice-based workflows, the approach aims to address incomplete anatomical coverage and operator dependence, and enables reformatting of reconstructed 3D volumes into arbitrary orientations for clinical interpretation. The study also introduces a new dataset of approximately 150 newborn hip ultrasound scans. Overall, the integrated acquisition–reconstruction–segmentation pipeline shows improved 3D segmentation performance relative to 2D baselines in the reported experiments.

**Strengths:**

- **Translational Relevance**: The paper presents an end-to-end pipeline that brings acquisition, 3D reconstruction, and downstream 2D/3D segmentation into a single framework, which is well aligned with real clinical workflows.
- **Study Design**: Leveraging 3D reconstruction to provide richer spatial context (compared to isolated 2D slices) is a compelling design choice that has the potential to improve robustness and consistency for clinical DDH assessment.
- **Dataset Contribution**: The authors contribute a new clinical DDH ultrasound dataset (~150 infant scans, collected across two sites/centers with multi-sweep acquisitions) and include useful cohort/demographic metadata, which strengthens transparency and supports reproducibility.
- **Clinical Relevance**: The clinical motivation and target application (e.g., infant DDH) are clearly articulated, making the problem setting and potential impact easy to follow.
- **Well-written Manuscript**: Overall, the manuscript is well written and thorough, with strong coverage of relevant prior work and supporting figures/tables (approach overview, data characteristics, and segmentation/visual examples).

**Weaknesses:**

- Although 3D reconstruction performs better, it is computationally expensive. What are memory usage and training times for the 2D and 3D segmentation and 3D reconstruction models? Could the authors also clarify how much scan/operator time increases when moving from single-slice 2D acquisition to multi-sweep 3D/volumetric coverage (e.g., added sweeps and total acquisition time)?
- Please describe some limitations of the proposed work.

**Detailed Comments:**

- The training parameter details in the appendices are helpful. Please also report memory usage and training times for the 2D and 3D segmentation and 3D reconstruction models.
- Please also include the number of trainable parameters for each segmentation and reconstruction model.
- In Figs. 7–9, the annotation text is hard to read. Increasing font size (and bolding) and improving figure resolution would improve readability.
- To show results variability across frames/subjects, please report standard deviation along with mean in Table 1.
- For reproducibility, authors should make the code and dataset publicly available. The contribution of a new dataset is especially useful if the authors make it public.

**Justification Of Final Rating:**

The authors have addressed my main concerns in the revised manuscript by adding computational details for both segmentation and reconstruction, quantifying the additional acquisition time required for the multi-sweep 3D protocol, and expanding the limitations discussion. They also improved figure readability and reported variability in the main results table. The paper is now substantially clearer and more reproducible.

**Justification Of The Preliminary Rating:**

The paper has sound validation-related strengths such as clear clinical and translational relevance, comprehensive evaluation, and contribution of a new dataset. However, it lacks computational details, and a discussion of limitations of the proposed work.

**Questions To Address In The Rebuttal:**

Address weaknesses:
- Please report memory usage and training times for the 2D and 3D segmentation and 3D reconstruction models.
- Please report scan/operator time for single-slice 2D acquisition and multi-sweep 3D/volumetric coverage.
- Please describe some limitations of the proposed work.

---

### Official Review · Reviewer_Avvt · 2026-01-10

**Confidence:** 5
**Preliminary Rating:** 5
**Final Rating:** 5

**Summary:**

This study presents a clinical validation of a 3D ultrasound reconstruction workflow for neonatal hip dysplasia screening to solve the limitations of standard 2D scans. The authors created a new real-world neonatal hip ultrasound dataset. The authors explored the segmetation abilities of 2D, 2.5D and 3D nnU-Net configurations. The results showed that the 3D models was more accurate than 2D models.

**Strengths:**

1.Creation of a real-world dataset of infant hip ultrasounds that allows for both standard 2D analysis and advanced 3D modeling.

2.The experiments are systematic, employing a logical progression from 2D to 3D validation.

**Weaknesses:**

The authors didn't metion if they compare the 2D and 3D models in the same domain (e.g., 2D area or 3D volume).
From the figure 4, the DSC is very unstable for CO-Border area and sometimes very low across the validation samples. It is improved in 3D model, but it may not the case after calculating the DSC in the interpolated 2D plane.

**Detailed Comments:**

1.For figure 10, you may want to add details for each subplots. I know they are 3D results but I don't know the relasionship between them.

2. Terms DSC is not defined.

**Justification Of Final Rating:**

The authors addressed my concerns by improving the 2D/3D visual comparison in Figure 3 and adding a fairer evaluation by reconstructing 2D outputs into 3D and comparing them to 3D ground truth. These changes make the results clearer and more convincing.

**Justification Of The Preliminary Rating:**

Overall, this is a strong paper. The experimental design follows a clear logic, and the results appear promising. However, the evaluation would be significantly more robust if the authors could clarify how the Dice Similarity Coefficient (DSC) was calculated for the 2D versus the 3D models to ensure a fair comparison.

**Questions To Address In The Rebuttal:**

1. Figure 3 currently limits its visualization to the 2D and 2.5D segmentation results. To better compare the 2D and 3D model results, would it be valuable to include a corresponding 2D slice interpolated from the final 3D segmentation volume with the same geometri parameters? If not, why?

2.Regarding Table 1, could you clarify if the reported DSC values represent slice-wise metrics for the 2D models versus volumetric metrics for the 3D models? If the evaluation domains differ (2D area vs. 3D volume), a direct comparison might be biased. Would it be more rigorous to standardize the evaluation? For instance, a way to do this is to interploate 2D slices from the 3D volumes to compare against the 2D ground truth?

---

### Author Rebuttal · Authors · 2026-01-25

**Rebuttal:**

We thank the reviewers for their valuable feedback, which helped improve the clarity and positioning of the manuscript. In response, we have revised the manuscript and provide detailed point-by-point answers below, referring to the relevant sections of the updated version.

**Supporting Material:**

/attachment/958d3c54560472dc5da1a0cd24ca5736656715fe.pdf

---

### Meta-Review · Area_Chair_jB59 · 2026-02-08

**Recommendation:** Accept (Oral)
**Confidence:** 5

**Metareview:**

This paper presents a well-designed and clinically relevant validation of a 3D ultrasound reconstruction and segmentation pipeline for neonatal hip dysplasia. Reviewers praised the end-to-end workflow, dataset contribution, and systematic 2D-to-3D evaluation.
Concerns regarding fair 2D/3D comparison, computational cost, reconstruction drift, limited initial 2D training data, and missing clinical metric validation were thoroughly addressed in the rebuttal through additional experiments, clearer evaluations, and expanded discussion of limitations.
Overall, the revisions sufficiently resolve the raised issues, and the remaining limitations are clearly acknowledged.

---

### Decision · Program_Chairs · 2026-02-14

Accept (Poster)